



# What CloudSat can't see: Liquid water content profiles inferred from MODIS and CALIOP observations

Richard M. Schulte[1], Matthew D. Lebsock[2], John M. Haynes[3]

[1]Department of Atmospheric Science, Colorado State University, Fort Collins, CO, USA
[2]NASA Jet Propulsion Laboratory, California Institute of Technology, Pasadena, CA, USA
[3]Cooperative Institute for Research in the Atmosphere, Fort Collins, CO, USA

*Correspondence to*: Richard M. Schulte (rick.schulte@colostate.edu)

**Abstract.** Single layer nonprecipitating warm clouds are integral to Earth's climate, and accurate estimates of cloud liquid
water content for these clouds are critical for constraining cloud models and understanding climate feedbacks. As the only
cloud-sensitive radar currently in space, CloudSat provides very important cloud profiling capabilities. However, a
significant fraction of clouds are missed by CloudSat, because they are either too thin or too close to the earth's surface. We
find that the CloudSat 2B-CWC-RVOD product misses about 73 % of nonprecipitating liquid cloudy pixels, and about 63 %
of total nonprecipitating liquid cloud water content, compared to coincident MODIS observations. Those percentages
increase to 84 % and 69 %, respectively, if MODIS "partly cloudy" pixels are included. We develop a method, based on
adiabatic parcel theory but modified to account for the fact that observed clouds are often subadiabatic, to estimate profiles
of cloud liquid water content based on MODIS observations of cloud top effective radius and cloud optical depth combined
with CALIPSO observations of cloud top height. We find that, for cloudy pixels that are detected by CloudSat, the resulting
subadiabatic profiles of cloud water are similar to what is retrieved from CloudSat. For cloudy pixels that are not detected by
CloudSat, the subadiabatic profiles can be used to supplement the CloudSat profiles, recovering much of the missing cloud
water and generating realistic-looking merged profiles of cloud water. Adding this missing cloud water to the CWC-RVOD
product increases the mean cloud liquid water path by 228 % for single layer nonprecipitating warm clouds. This method
will be included in a subsequent reprocessing of the 2B-CWC-RVOD algorithm.

## 1 Introduction

Liquid clouds are a key part of the climate system. They have important influences on Earth's radiative balance (Hartmann et
al., 1992), the hydrological cycle, and on local and large-scale circulations (e.g., Ma et al., 1996). Low clouds reflect a large
amount of incoming sunlight, without changing the amount of outgoing longwave radiation by very much. Because of this,
changes in the extent and properties of low clouds have important climate feedback implications, and the representation of
clouds has long been recognized as one of the most significant sources of uncertainty in global climate models (e.g., Cess et
al., 1989; Stephens et al., 2010; Zelinka et al., 2016). Accurate estimates of liquid cloud water are needed for evaluating and



improving cloud models. Satellite datasets are well-suited to this purpose because they can provide near-global coverage using a consistent instrument.

To date, the only global-scale observations of the vertical profiles of liquid water content (LWC) of low-altitude liquid clouds are derived from the CloudSat satellite (Stephens et al., 2008), carrying the 94-GHZ Cloud Profiling Radar

(CPR; Tanelli et al., 2008). These profiles provide utility for process studies, validation of model output, and as input for forward radiative transfer calculations of shortwave and longwave radiative heating profiles. For much of its lifetime (until 2018), CloudSat was part of the A-Train constellation of satellites, a constellation that includes several other instruments capable of measuring cloud properties. For example, one of the instruments on the *Aqua* satellite is the Moderate Resolution Imaging Spectroradiometer (MODIS), which can passively measure various cloud optical properties (King et al., 1992).

Meanwhile the Cloud-Aerosol Lidar with Orthogonal Polarization (CALIOP) on the *CALIPSO* satellite (Winker et al., 2009) can detect the presence and cloud top height of even very thin clouds, although its signal rapidly attenuates and so it does not possess the same profiling capabilities of CPR. Passive microwave radiometers, such as the Advanced Microwave Scanning Radiometer for EOS (AMSR-E) which is also on *Aqua*, are commonly used to estimate LWP; however, these instruments have large footprints that complicate direct, pixel-level comparisons with CloudSat and are sensitive to the liquid water path

which is difficult to decompose into cloud and precipitation components (Lebsock and Su, 2014).

The CloudSat Radar-Visible Optical Depth Cloud Water Content Product, 2B-CWC-RVOD (Leinonen et al., 2016) provides CPR-based profiles of LWC, with MODIS observations of optical depth used as an additional constraint. One shortcoming of the 2B-CWC-RVOD algorithm (hereafter simply RVOD) is that it fails to account for clouds that are not detected by CPR. This can happen for two reasons. First, CPR's surface clutter extends up to the third range bin above the

surface, or about 750-1000 m (Tanelli et al., 2008). Clouds that are mostly or entirely in this part of the vertical column will be masked by clutter from the bright radar surface backscatter. Second, CPR has a minimum detectable reflectivity of about –30 dBZ. If the cloud droplets are not numerous or large enough to generate reflectivities of this magnitude (after averaging to the CPR range resolution), then RVOD will not generate LWC estimates for that cloud. The combination of these two effects results in a sampling bias, where the RVOD algorithm is weighted towards the thickest and highest liquid water

content clouds. While it is not the main focus of our study, RVOD also has two important biases when precipitation is present. First, drizzle and raindrops are much more reflective than cloud drops so that the RVOD algorithm cannot accurately assign a cloud water content in the presence of precipitation. Second the algorithm cannot determine where the cloud base is so it assigns cloud water content to range bins that are beneath the cloud base where precipitation results in strong reflectivity values.

Is has been noted (e.g., Christensen et al., 2013; Li et al., 2018) that CloudSat misses a non-trivial percentage of clouds, either because the clouds are within the radar's surface clutter zone or because the reflectivities are below the radar threshold. This study attempts to quantify how much is missed, and then develops a method to "fill in" much of the missing cloud water mass using coincident observations from MODIS and CALIOP, which are more sensitive to these thin clouds. Our focus in this study is on nonprecipitating, single layer warm liquid clouds. While there are many other types of clouds



that are observed by CloudSat, this type of cloud is the most easily modelled by our proposed combination of CALIOP cloud
top plus MODIS optical depth and effective radius. In addition, as mentioned above, these types of clouds have profound
effects on Earth's radiation budget. In section 2, we describe the data sources used in our study. In section 3, we develop a
subadiabatic cloud model that we use to make LWC profile estimates where RVOD estimates are not available. In section 4,
we calculate how much total cloud water from these types of clouds is missed by RVOD compared to MODIS and CALIOP,
and in section 5, we evaluate the performance of the subadiabatic model. Section 6 contains our conclusions.

## 2 Data

The data used in this study come from 3 main instruments: CPR, MODIS, and CALIOP. The specific data products used are
detailed below. For our analysis, we consider 10 years of data, from 2007-2016, but only use days when all three instruments
have valid data. This results in a total of 489,364,826 CloudSat profiles. In all cases, we use the R05 version of each product
as obtained from the CloudSat Data Processing Center (DPC; https://www.cloudsat.cira.colostate.edu).

## 2.1 CloudSat

The CPR onboard CloudSat is a 94-GHz nadir-pointing radar which measures the amount of microwave radiation
backscattered by hydrometeors as a function of distance between the satellite and the earth's surface (Stephens et al., 2018).
It has a minimum detectable reflectivity factor of about -30 dBZ, cross-track resolution of 1.4 km, along-track resolution of
1.7 km, and vertical resolution of 480 m (sampled every 240 m). We use radar reflectivities and geolocation variables from
the 2B-GEOPROF product (Marchand et al., 2008), and estimates of liquid water content and cloud droplet effective radius
from the RVOD algorithm.

RVOD retrieves cloud water and ice contents from CPR radar reflectivity observations combined with MODIS
cloud optical depths. Since MODIS cloud optical depth is only available during the daytime, RVOD is a daytime-only
product. RVOD is based on the optimal estimation framework (Rodgers, 2000). For each CloudSat pixel, the algorithm seeks
to retrieve a profile of hydrometeors that is consistent with the observed CPR profile of radar reflectivity, the MODIS optical
depth, and a priori assumptions. See Leinonen (2016) for full details about the algorithm; here we review only a couple
assumptions that are relevant to this work.

First, note that RVOD only retrieves a CWC value where the 2B-GEOPROF product indicates a cloud is present,
even if the MODIS optical depth is greater than zero (indicating the likely presence of a cloud). In these cases, it is likely
that there is a cloud, but that the cloud is either too low to be seen by CloudSat (it is hidden by radar surface clutter), too thin
to be seen by CloudSat (the cloud droplets do not produce reflectivities above -30 dBZ), or both. Second, note that RVOD
assumes that liquid cloud droplets follow a lognormal size distribution

$$N(r) = \frac{N_T}{\sqrt{2\pi}\sigma_{log}r}\exp\left[\frac{-\ln^2(r/r_g)}{2\sigma_{log}^2}\right], \tag{1}$$





where $r$ is the drop radius, $r_g$ is the geometric mean radius, $N_T$ is the total number concentration (assumed constant throughout the column for liquid clouds), and $\sigma_{log}$ is the scale parameter, fixed at 0.38 based on Miles et al. (2000). For a lognormal size distribution, $r_g$ is related to the effective radius, $r_e$, by Eq. (2):

$$r_e = r_g \exp\left(2.5\sigma_{log}^2\right). \tag{2}$$

We use Eq. (2) to convert from the $r_g$ reported in the RVOD files to the $r_e$ that we compare against MODIS $r_e$ in Section 4.

## 2.2 MODIS

The MODIS instrument is a spectroradiometer that captures data in 36 spectral bands ranging in wavelength from 0.4 um to 14.4 um and at spatial resolutions ranging from 250 m to 1 km (Justice et al., 1998). We use MODIS data from the instrument onboard the Aqua satellite, which flew in formation with CloudSat as part of the A-train during the period of this study, providing near-coincident observations of clouds. Specifically, we make use of the MODIS-1KM-AUX product
produced by the CloudSat Data Processsing Center (DPC). This dataset contains a subset of MODIS MYD06 retrieved cloud properties that are collocated with each CPR footprint. Data are provided at 1 km resolution, and we use the 1 km MODIS pixel whose center is closest to the center of the CPR footprint for each matchup.

        MODIS retrieves cloud effective radius and cloud optical thickness simultaneously using the bispectral technique ( Nakajima and King, 1990; Platnick et al., 2003). In this method, a water absorbing band is combined with a nonabsorbing
band (either 0.65, 0.85, or 1.2 $\mu m$, depending on the surface type). There are three versions of the retrieval using either the 1.6, 2.1, or 3.7 $\mu m$ MODIS channel as the absorbing channel in the bispectral calculation. For our standard analysis, we choose to use the 3.7 $\mu m$ version, as this channel is the most strongly absorbing and thus the most sensitive to cloud top properties (Platnick et al., 2003). However, we also test the effects of using the other versions in Sect. 5. MODIS-1KM-AUX also flags some pixels as being "partly cloudy" (hereafter referred to as PCL pixels), and gives retrieved $r_e$ and $\tau$ for those
pixels as well. In most cases, we include PCL pixels in our analyses, although in some cases we test the effect of withholding them.

### 2.3 CALIOP

The Cloud-Aerosol Lidar with Orthogonal Polarization (CALIOP) onboard the CALIPSO satellite is a two-wavelength (532 nm and 1064 nm) polarization-sensitive lidar that provides vertical profiles of aerosols and clouds at 333 m horizontal
resolution and 30 m vertical resolution (Hunt et al., 2009). In our study, we use the 2B-CLDCLASS-LIDAR product from the CloudSat DPC, which combines and collocates CPR and CALIOP measurements for the purposes for classifying clouds (Sassen et al., 2008). We use the variable "Cloud Layer" to screen for single layer clouds, the variable "Cloud Layer Top" to determine the cloud top height, and the variable "Cloud Phase" to determine whether a cloud is liquid phase.



## 2.4 Auxiliary information

In order to use the subadiabatic model described in Sect. 3.1 to estimate cloud LWCs, we must assume a temperature and pressure for the cloud. We use data from the European Centre for Medium-Range Weather Forecasts (ECMWF) HRES (high resolution) forecast model. This data is collocated to the CPR profiles and provided in the DPC's ECMWF-AUX product. From ECMWF-AUX we take the temperature and pressure at cloud top, as identified by 2B-CLDCLASS-LIDAR, use these values as input to the subadiabatic cloud model described in Sect. 3.1 and Appendix A.

## 130 3 Methods

As mentioned in the introduction, this study is concerned exclusively with single layer, nonprecipitating warm liquid clouds. To be classified as nonprecipitating, a given CloudSat pixel must not have any reflectivities above -15 dBZ anywhere in the column (not including surface clutter). This threshold is similar to the threshold used for the rain flag from 2C-PRECIP-COLUMN (Haynes et al., 2009). That algorithm uses a "near-surface" reflectivity threshold of -15 dBZ, after accounting for 135 attenuation. We use 2B-CLDCLASS-LIDAR to identify pixels that have exactly one cloud layer, are liquid phase, and have a cloud top below 5 km.

Additionally, we screen out pixels which have cloud top temperatures colder than 273 K (according to the temperature profile from ECMWF-AUX and the cloud top height from 2B-CLDCLASS-LIDAR). This is done because RVOD classifies these clouds as mixed phase, even though the CALIOP observations suggest that the clouds are frequently 140 composed of supercooled liquid. This scenario proves problematic for the RVOD algorithm. RVOD was primarily designed with liquid clouds in mind, and for any cloud colder than 273 K, the algorithm artificially partitions the total water path to be a mixture of liquid and ice that depends on temperature. The ice estimate comes from the 2C-ICE algorithm, and the result in many cases is an unrealistically low retrieved value of $r_e$ for the water droplets. This is demonstrated in Fig. 1. This deficiency will be addressed in future releases of the RVOD by forcing the algorithm to assume liquid cloud droplets when 145 CALIOP indicates the cloud is liquid phase. Here, we simply restrict our analysis to warm clouds that are unequivocally liquid. After all screening, we are left with about 10.8 % of all CloudSat pixels that are classified as nonprecipitating, single layer warm liquid clouds. This percentage is highly variable regionally, as explored further in Sect. 4 (and Fig. 3).

### 3.1 Sub-adiabatic cloud model

Where RVOD does not retrieve any cloud water, but CALIOP and MODIS both indicate the presence of a cloud, we can be 150 fairly confident that such a cloud exists, but that it is either too thin or too low to be detected by the CPR. The general idea of our scheme is to use the MODIS measurements to estimate how much total cloud water is present in the column, and then use the cloud top height from CALIOP combined with some assumptions about vertical structure to apportion the cloud water in the vertical. There are two classical approaches to this problem. The first is to assume that LWC is vertically homogenous (e.g., Stephens, 1978). The second is to assume that cloud water linearly increases from base to cloud top,





while cloud droplet number concentration stays constant (e.g., Brenguier et al., 2003). This assumption corresponds to adiabatic growth of cloud droplets as they are lifted through the saturated air of the cloud. Both assumptions are convenient as they lead to tidy expressions for the LWP of a cloud as a function of cloud top droplet effective radius $r_e$ and cloud optical depth $\tau$. These expressions have the form

$$LWP = \gamma \rho_l r_e \tau, \tag{3}$$

where $\rho_l$ is the density of liquid water and $\gamma = 2/3$ for the vertically uniform cloud and $\gamma = 5/9$ for the adiabatically stratified cloud (Wood and Hartmann, 2006).

However, field studies have shown that real liquid clouds tend to fall somewhere between these two assumptions. They have LWCs that do increase from cloud base towards cloud top, but the rate of increase is less than that predicted by the adiabatic model (e.g., Brenguier et al., 2000; Rangno and Hobbs, 2005; Rauber et al., 2007; Min et al., 2012). Boers et al.

(2006) was one of the first studies to lay out a framework for modelling the subadiabaticity of a cloud as a function of cloud depth. We use a simpler adjustment to the adiabatic model, first proposed by Wood et al. (2009), that is meant to account for entrainment, mixing, and other processes that tend to give actual clouds sub-adiabatic growth rates. In a fully adiabatic model, the LWC $l$ of a cloud would vary with height $h$ above cloud base according to Eq. (4):

$$l(h) = c(T,P)h, \tag{4}$$

where $c(T,P)$ is the moist adiabatic condensation rate at temperature $T$ and pressure $P$, given by Eq. (5):

$$c(T,P) = \rho_{air} \frac{c_p}{L_v}(\Gamma_d - \Gamma_m). \tag{5}$$

Here is $\rho_{air}$ is the air density of a fully saturated air parcel at temperature $T$ and pressure $P$, $c_p = 1004$ J/kg K is the specific heat of dry air at constant pressure, $L_v = 2.26$ x $10^6$ J/K is latent heat of vaporization of water, $\Gamma_d = 9.8$ K/km is the dry adiabatic lapse rate, and $\Gamma_m$ is the moist adiabatic lapse rate at $T$ and $P$. Wood (2009) modifies Eq. (4) by introducing a

scaling factor, $z_0$:

$$l(h) = c(T,P)h\frac{z_0}{z_0+h}. \tag{6}$$

With this formulation, shallow clouds tend to be closer to adiabatic than deeper ones. We use Eq. (6) in our subadiabatic model. In most cases we assume $z_0 = 500$ m, following *in situ* data from Ragno and Hobbes (2005), although we do test other values of $z_0$ in Sect. 5.

The liquid water content of a cloud droplet size distribution given by $n(r)$ is defined as

$$l = \frac{4}{3}\pi\rho_l \int r^3 n(r)dr. \tag{7}$$

The effective radius is defined as



$$r_e = \frac{\int r^3 n(r)dr}{\int r^2 n(r)dr}. \tag{8}$$

The extinction coefficient is given by

$$k_{ext} = \int Q_{ext} \pi r^2 n(r)dr, \tag{9}$$

where $Q_{ext}$ is the extinction efficiency. Combining Eqs. (7-9) yields

$$k_{ext} = \frac{3Q_{ext}l}{4\rho_l r_e}. \tag{10}$$

The optical depth is the integral of this equation over the cloud depth H:

$$\tau = \frac{3Q_{ext}}{4\rho_l} \int_0^H \frac{l}{r_e} dh. \tag{11}$$

If one has estimates of $\tau$ and cloud top $r_e$, then one can use Eqs. (6), (7), (8), and (11) to solve for $H$ and the profile of $l(h)$. The details of the inversion are given in Appendix A. It is using this procedure that we convert MODIS estimates of $\tau$ and $r_e$ into a modelled profile of cloud liquid water. The LWP is then the integral of $l(h)$ over the cloud depth. Figure 2 shows a comparison of the subadiabatic, adiabatic, and vertically homogenous methods of distributing cloud water, for a cloud with a cloud top height of 1500 m, a cloud top $r_e$ of 15 $\mu$m, and an optical depth of 29. It can be seen that, compared to the fully adiabatic model, the subadiabatic model yields clouds that are slightly deeper, with vertical gradients in LWC that are more gradual near the top of the cloud, lower maximum LWCs, and lower cloud droplet number concentrations.

The final step in creating subadiabatic profiles of LWC for comparison against RVOD is to average the resulting profiles to the resolution of CPR. To accomplish this, once we have solved for a profile of $l(h)$ as described above, that profile is run through a Gaussian-weighted moving average filter. The filter has a 6 dB window size of 480 m, corresponding to CPR's range resolution. The filtered profile is then sampled every 240 m, at the center of each CPR bin.

## 4 Comparisons between A-Train estimates of liquid cloud water

In this section we compare estimates of single layer nonprecipitating warm (SLNPW) clouds from CPR, MODIS, and CALIOP. We first consider estimates of cloud frequency, and then consider estimates of the total amount of water present in these clouds.

### 4.1 Cloud Frequency

Let us first quantify retrieval cloud fractions from the RVOD and MODIS cloud optical properties algorithms. We note that MODIS in particular reports a significant number of cloudy pixels with no associated cloud properties from the optical properties algorithm. It is clear when looking at the co-located data that CloudSat and MODIS both fail to report cloud properties for a significant portion of the SLNPW clouds that are seen by CALIOP, but also that MODIS captures many more of these clouds than does CloudSat. This can be seen in Figure 3, which shows a map of the SLNPW cloud fraction





from each of these satellites. From all of the maps, it is clear that the SLNPW cloud fraction is greatest (exceeding 50 %, according to the lidar data) in the subtropical areas to the west of the continents. These are areas known for commonly having extensive stratocumulus cloud decks (Klein and Hartmann, 1993). According to RVOD, these areas only have SLNPW clouds 10-15 % of the time, while MODIS reports cloud properties closer to 25 % of the time, or up to 35 % of the

time if partly cloudy (PCL) pixels are included. Outside of these areas where stratocumulus clouds are common, the detection percentages are even worse. Overall, for all SLNPW clouds detected by CALIOP, only 6 % are detected by RVOD, 22 % detected by MODIS, and 37 % detected by MODIS if PCL pixels are included.

Why are these CALIOP-detected clouds being missed by CloudSat? One possibility that we considered was that RVOD was missing these clouds because they are too close to the surface, and thus masked by surface clutter in the CPR

reflectivities. However, Figure 4 demonstrates that this is only part of the explanation. This figure shows the fraction of lidar-detected SLNPW clouds that are detected by CloudSat and MODIS, as a function of cloud top height. Sure enough, clouds with tops below 1 km are almost never detected by CloudSat, but even clouds with higher tops have detection percentages only between 10-20 %. Meanwhile, MODIS detects these clouds around a third of the time (or near 40 % including PCL pixels), without too much of a dependence on cloud top height. The higher MODIS detection percentages for

cloud top heights below 500 m are likely an artifact of the small sample size of clouds that are that shallow, while the small bump in detection percentages at cloud top heights near 1500 m is likely due to the fact that this is around the typical depth of the boundary layer over the oceans, where a lot of the thickest SLNPW clouds tend to top out.

**4.2 How much cloud water is missed by RVOD?**

From the previous section, it is clear that RVOD fails to detect a majority of SLNPW clouds. However, the clouds that are

missed are likely to be particularly thin, since they do not generate large enough radar reflectivities to be seen by CPR. Is the total *amount* of liquid cloud water that is missed by RVOD significant? Once again, the answer is yes. The average SLNPW cloud LWP, averaged over all CloudSat observations (that is, including observations where there is no cloud or the cloud is not a SLNPW cloud), is only 5.4 g m$^{-2}$ for RVOD, compared to 14.4 g m$^{-2}$ for MODIS (retrieved using our subadiabatic model) or 17.5 g m$^{-2}$ for MODIS including PCL pixels. Figure 5 shows the cumulative distribution of these averages as a

function of cloud top height. Once again it is clear that, while a significant portion of the missing cloud water comes from clouds with tops below 1 km, there is a large gap between RVOD and MODIS cloud water even for cloud top heights above this level (i.e., the lines in Figure 5 continue to diverge). Figure 6 plots the spatial distribution of the missing cloud water. Unsurprisingly, the areas of the world where RVOD misses the most SLNPW cloud water overlap heavily with where SLNPW cloud fractions are highest. It should be noted that these estimates of estimates of missing cloud water do not

include the water in the clouds that CALIOP sees but MODIS does not, as we do not have independent estimates of LWP for these clouds that are detected only by CALIOP.



## 5 Augmenting CloudSat LWC profiles using information from MODIS and CALIOP

We have shown that the CloudSat radar (in particular, the RVOD retrieval algorithm) misses a lot of liquid cloud water that is seen by MODIS. It is desirable to augment the RVOD profiles of cloud water with estimates from MODIS in areas where MODIS detects a cloud but RVOD doesn't. These MODIS-derived profiles are likely to be less accurate than the CloudSat derived ones (because they are not constrained by radar observations), but are nonetheless much more useful for model evaluation than assuming that all of these areas of thin clouds and completely free of cloud water, as RVOD currently does.

We first test the reliability of the subadiabatic cloud model by constructing profiles of LWC for all SLNPW pixels in our dataset that are seen by RVOD. The profiles are generated using MODIS cloud top $r_e$ and $\tau$ and CALIOP cloud top height, according to the procedure described in Section 3. Then we compare these subadiabatic estimates of liquid cloud water against the RVOD estimates for the same pixels. Figure 7 shows RVOD estimates of LWP, column-maximum LWC, and cloud top $r_e$ alongside the corresponding estimates from the subadiabatic model. There is good agreement between the two methods, especially when it comes to the integrated LWP estimate. Note that the LWP agreement is better for the subadiabatic model than it would be if we used the standard LWP estimates included in the MODIS-1KM-AUX data files, as these assume a vertically uniform profile of cloud water. There is decent agreement when it comes to the profiles of LWC as well; however, the subadiabatic model tends to create clouds that are slightly less thick than the RVOD profiles suggest. A sign of this is seen in the distributions of column-maximum LWC shown in Fig. 7. The maximum LWC from the subadiabatic model tends to be slightly higher than the maximum from RVOD, indicative of a thinner cloud with a sharper gradient in LWC. Nevertheless, the modelled clouds are still thicker than they would be if we used the adiabatic model to distribute cloud water. Finally, the MODIS $r_e$ estimates tend to be a bit larger than the estimates of RVOD, a finding that has been reported in other studies as well (Zhang and Platnick, 2011; Painemal and Zuidema, 2011).

### 5.1 Case Studies

Two case studies illustrate the usefulness and potential shortfalls of the subadiabatic model for filling in profiles of LWC in areas where RVOD misses clouds. The first case is shown in Figure 8, and comes from 1 February 2007, when the A-train observed a deck of low clouds off the west coast of Chile. According to our screening criteria, the entire segment of observations shown in Fig. 8 consists of single layer nonprecipitating warm clouds, with CALIOP cloud top heights between 1-2 km. The fact that clouds are present are confirmed by the 11 $\mu$m MODIS channel infrared brightness temperatures shown in panel (a). The subadiabatic model yields LWP estimates that are very similar to RVOD for the pixels that CloudSat sees, as seen in panel (b). However, RVOD misses over half of the cloudy pixels. Most of these missed pixels have LWPs (as determined by the subadiabatic model) smaller than around 75 g m$^{-2}$. Panels (c) and (d) show the profiles of retrieved LWC from both RVOD and the subadiabatic model. For the clouds that are seen by CloudSat, the cloud depths from the subadiabatic model (in terms of number of radar bins) are similar to the cloud depths from RVOD. However, the modelled liquid water content tends to be slightly more concentrated in the top half of the cloud for the subadiabatic model. Perhaps



the biggest benefit of the subadiabatic model estimates is that it allows us to created merged LWC profiles, as demonstrated
in panel (e). In the merged model we use the RVOD-estimated profile of LWC where available but, for columns that have no
RVOD retrieval, use the estimate from the subadiabatic model instead. For this case, the merging process creates a smooth
and very plausible-looking thin layer of liquid cloud water, with no sharp discontinuities at the edges of the clouds that are
thick enough to be detected by RVOD.

The second case comes from the Indian Ocean on 2 January 2007, and is shown in Figure 9. This case includes
clouds that are slightly higher in altitude, and includes some pixels that are flagged as precipitating. Once again, as indicated
by the infrared brightness temperatures, this entire scene consists of clouds, but RVOD detects only about half of the cloudy
pixels. For the thinner, nonprecipitating clouds, there is good agreement between the RVOD and subadiabatic model
estimates of liquid water path (for the clouds that are detected by RVOD). However, where rain is present, the subadiabatic
model yields lower estimates of LWP than RVOD. This is not surprising, as the radar reflectivity is dominated by the larger
precipitation drops, whereas the MODIS observations are primarily sensitive to the smaller cloud drops. For a precipitating
cloud, the radar reflectivity will be maximized lower in the column, as the larger drops grow by coalescence and precipitate
out of the base of the cloud. The cloud depths from RVOD are significantly complicated by the fact that precipitation
particles dominate the CPR radar return, as discussed in the introduction, and are likely too thick. The merged LWC model,
which in this case uses the subadiabatic model estimate for pixels flagged as precipitating as well as pixels with no cloud
retrieved from RVOD, still performs well in filling in the gaps between CloudSat-detected clouds. However, there are some
are some discontinuities in the cloud thickness for the precipitating regions, which probably represents residual influence
from drizzle drops that do not quite meet our -15 dBZ threshold. In the future, we plan to transition precipitating pixels'
cloud water content to the subadiabatic model and have the radar derive the precipitation water content following Lebsock
and L'Ecuyer (2011). For the time being we emphasize that pixels that are identified as precipitating are likely to have water
contents which are too high.

## 5.2 VOCALS Cross Section Analysis

Next, we explore the performance of RVOD and the subadiabatic model in detecting and estimating profiles of liquid cloud
water in an area of the world dominated by a persistent marine stratocumulus cloud deck. The VAMOS Ocean-Cloud-
Atmosphere-Land Study (VOCALS) was an international research program focused on the improved understanding and
modelling of the southeastern Pacific climate system (Mechoso et al., 2014). As part of VOCALS, instrumented moorings
were installed near 20° S, 85° W and 20° S, 75° W (Colbo and Weller, 2007, 2009). Many previous studies of stratocumulus
clouds in this region have thus focused on the 20° S parallel (e.g., Serpetzoglou et al., 2008; Zuidema et al., 2009). For our
purposes, we cut a cross section centered on 20° S from 90° W to 70° W and call this the "VOCALS" cross section. All
A-train observations included in our dataset and within ±5° of 20° S are binned in two degrees wide longitude bins for
plotting in Figures 10 and 11.



Figure 10 shows the fraction of the time each CPR bin along the VOCALS cross section contains a single layer nonprecipitating warm cloud, as detected by CALIOP, RVOD, or the merged model. As before, the merged model uses the RVOD LWC profiles for clouds detected by RVOD, but adds the subadiabatic profiles for clouds not seen by RVOD. In some bins, the SLNPW cloud fraction from CALIOP is near 80 %, reiterating just how prevalent those types of clouds are in this area. The CALIOP plot also shows that the cloud heights tends to increase as one moves further away from the coast (i.e., east to west), consistent with the growth of the marine boundary layer as it advects over warmer sea surface temperatures (Krueger et al., 1995). RVOD alone detects only a small portion of these SLNPW clouds, and it particularly struggles to detect clouds close to the coast, which tend to be lower and thinner. Using the merged model greatly improves SLNPW cloud detection, even though a lot of clouds that are seen by CALIOP are still missed. The gains are especially striking below 1 km and in the eastern part of the domain.

Figure 11 shows a similar series of plots for the VOCALS cross section, but looking at SLNPW cloud liquid water content instead of cloud fraction. Because CALIOP alone does not give LWC estimates, only estimates from RVOD and the merged model are compared. The average cloud liquid water contents from the merged model are between 0.01 and 0.02 g m$^{-3}$ larger than from the RVOD algorithm. Note that those values come from averaging over all pixels, not just pixels that contain SLNPW clouds. Similar to Fig. 10, we see that the largest differences between the merged model and RVOD occur closer to the coast, where the clouds tend to be lower.

**5.3 Sensitivities and Uncertainties**

The subadiabatic LWC profile derived from MODIS observations of $r_e$ and $\tau$ depends both upon which MODIS absorbing channel is used in the bispectral technique, and the value of the scaling factor, $z_0$, used to describe the shape of the vertical profile. To explore the effect of these choices, we calculated SLNPW cloud LWC profiles from the year 2016 using nine different combinations of MODIS channel and scaling factor. Specifically, we tested using the 1.6, 2.1, and 3.7 $\mu$m MODIS channels with $z_0$ equal to either 100, 250, or 500 m. The mean and standard deviation of several relevant derived cloud parameters for each of the nine experiments are given in Table 1. The 1.6 $\mu$m MODIS channel misses about 50 % of the SLNPW clouds that are detected by the 2.1 and 3.7 $\mu$m channels. This is mostly due to the fact that several of the 1.6 $\mu$m MODIS detectors on *Aqua* are inoperable (K. Meyer, personal communication, 2023). For pixels which are detected by all MODIS channels, using a smaller wavelength channel tends to give wider distributions of $r_e$ and N, and a slightly larger LWP, on average. For $z_0$, using smaller values gives larger cloud depths, smaller number concentrations, slightly higher liquid water paths, and lower maximums in LWC. The effect is much more pronounced when comparing $z_0 = 100$ m to $z_0 = 250$ m than when comparing $z_0 = 250$ m to $z_0 = 500$ m.

We can use these sensitivity tests to generate a crude estimate of the uncertainty inherent in the cloud liquid water paths derived from the subadiabatic model. We define the fractional uncertainty of each cloudy pixel using Eq. (12):

$$\epsilon = \frac{W_{max} - W_{min}}{W_{best}}, \tag{12}$$



where $W_{max}$ is the maximum LWP estimated from the 9 sensitivity experiments, $W_{min}$ is the minimum LWP, and $W_{best}$ is our best estimate, defined to be the estimate of the LWP using the 3.7 $\mu$m MODIS channel and $z_0 = 500$ m (i.e., the version

used in the rest of this paper). The median fractional uncertainty in the LWP for all SLNPW cloudy pixels is 0.38, with a 25th percentile value of 0.214 and a 75th percentile value of 0.666. Figure 12 shows that this uncertainty tends to be smallest (typically less than 0.3) in the areas of the world where the single layer nonprecipitating warm clouds are most prevalent. This represents the typical uncertainty in the LWP retrieved at each *pixel*; the uncertainty in the mean LWP is considerably less. As Table 1 shows, the lowest estimate for the mean SLNPW LWP from the 9 experiments is 53 g m$^{-2}$ while the highest

estimate is 62 g m$^{-3}$, a fractional uncertainty of about 0.17.

**6 Conclusion**

Single layer, nonprecipitating warm clouds make up about 11 % of all A-train pixels in our dataset, including a prevalence above 75 % over key areas of the globe dominated by stratocumulus cloud decks. Given the radiative importance of these clouds, it is troubling that the current RVOD product fails to detect many of these clouds. Globally, our analysis indicates

that greater than 90 % of all CloudSat pixels which at least partially contain a single layer nonprecipitating warm cloud (according to CALIOP) have no cloud water content in the RVOD product. Performance is better, but still problematic, over the stratocumulus cloud decks. While MODIS also misses many of these thin clouds, it finds about 6 times as many CloudSat pixels containing SLNPW clouds (if PCL pixels are included).

This creates an opportunity to leverage coincident daytime MODIS observations in order to augment RVOD

estimates of SLNPW cloud water. While it is common to use MODIS estimates of $r_e$ and $\tau$ to estimate cloud liquid water path, this study is novel in the way that *profiles* of cloud liquid water content are generated. Instead of assuming a vertically homogeneous or adiabatically stratified cloud, we assume a subadiabatic cloud model that, when combined with CALIOP estimates of cloud top height, generates a full LWC profile. That said, our method still produces estimates of liquid water path. One of the more striking results of the study is that, when considering only cloudy pixels that are seen by RVOD, the

RVOD and subadiabatic model estimates of LWP agree extremely well. This allows us to feel more confident in extending the method to produce profiles of LWC for clouds that are too low or too thin to be detected by RVOD and to a lesser extent to clouds which are precipitating. The case studies that we have presented indicate that when we merge the two methods; i.e., the heritage RVOD algorithm combined with the subadiabatic model for pixels where RVOD does not see a cloud, we obtain smooth and realistic-looking curtains of cloud liquid water content.

We intend to include these merged LWC profiles in the next reprocessing of RVOD that will be produced when the full CloudSat dataset is reprocessed at the end of the mission towards the end of 2023. These merged profiles could be useful for other products as well, such as the CloudSat FLXHR-Lidar product (Henderson et al., 2013). This product currently uses RVOD profiles of LWC where available but uses climatological averages for LWC and $r_e$ where CALIOP detects a cloud but RVOD does not. This study provides a better method for assigning LWC based on actual MODIS observations and

providing vertically resolved inputs including a physically plausible cloud base to the radiative transfer model.





There are several limitations to our method that must be mentioned. For one, it does not account for clouds that are missed even by MODIS. In these cases, it may be possible to use the attenuated backscatter and/or the path integrated attenuation from CALIOP to constrain the cloud optical depth and with the assumption of a cloud effective radius the method might be extended to more pixels. Since this method relies upon measurements at near-visible wavelengths from
MODIS, it can only be used during the daytime (this is also a limitation of the existing RVOD algorithm). We have also not considered precipitating clouds, multi-layered clouds, or clouds with ice in them. These types of clouds all create different kinds of uncertainties for LWC retrievals. It is also worth noting that the inversion method derived here, which includes a vertically varying subadiabaticity, influences the derived cloud droplet number concentration in addition to the LWC profile. This sensitivity deserves future study in its own right. There are a large number of papers that use an adiabatic or
subadiabatic model to derive cloud droplet number (see Grosvenor et al., 2018, and references therein).

Finally, this method of partitioning cloud LWC will be relevant to future cloud observing satellite missions such as EarthCARE (Illingworth et al., 2015) and NASA's planned Atmosphere Observing System (AOS). Both missions will combine a cloud-sensitive radar with a lidar and MODIS-like instruments. EarthCARE's radar is projected to have better sensitivity (-35 dBZ) than CPR, but will still likely miss some thin and/or low liquid clouds. Meanwhile AOS's sensitivity is
still being determined but is likely to be less sensitive than CPR. In both cases, the method presented here could be used to supplement LWC profile estimates.

**Appendix A**

Here we describe how we invert MODIS estimates of $\tau$ and cloud top $r_e$ to obtain a profile of cloud liquid water content, $l(h)$, using the assumptions of the sub-adiabatic model outlined in Sect. 3.1. As derived in Martin et al. (1994) and
elsewhere, the effective radius of a droplet distribution can be related to the liquid water content $l$ and total droplet number concentration $N$ (assumed to be constant throughout the cloud) by Eq. (A1):

$$r_e^3 = \frac{l(h)}{4/_3 \pi \rho_l k N} , \tag{A1}$$

where k relates the effective radius to the volume mean radius ($k = \frac{r_v^3}{r_e^3}$) and is assumed to be equal to 0.8 in accordance with Grosvenor et al. (2018). Using the expression for $l(h)$ given in Eq. (6), and evaluating at cloud top, we arrive at Eq. (A2):

$$r_e(H) = \left[ \frac{3 z_0 c H}{4 \pi \rho_l k N (z_0 + H)} \right]^{1/3} . \tag{A2}$$

Meanwhile, substituting our expressions for $l(h)$ and $r_e$ given in Eqs. (6) and (A1), respectively, into the expression for $\tau$ given in Eq. (11) yields the relation

$$\tau = \left( \frac{3c}{4\rho_l} \right)^{2/3} Q_{ext} \pi^{1/3} (kN)^{1/3} \int \left( \frac{z_0}{z_0+h} h \right)^{2/3} dh . \tag{A3}$$



For positive $h$ and $z_0$, the integral on the right hand side of Eq. (A3) is given by

$$\int \left(\frac{z_0}{z_0+h}h\right)^{2/3} dh = \frac{3}{5}h^{5/3} {}_2F_1\left(\frac{2}{3},\frac{5}{3},\frac{8}{3},-\frac{h}{z_o}\right), \tag{A4}$$

where ${}_2F_1$ is the hypergeometric function. Substituting Eq. (A4) into Eq. (A3) and evaluating from cloud base to cloud top gives the following expression for optical depth:

$$\tau = \frac{3Q_{ext}}{5}\left(\frac{3c}{4\rho_l}\right)^{2/3}(k\pi N)^{1/3}H^{5/3} {}_2F_1\left(\frac{2}{3},\frac{5}{3},\frac{8}{3},-\frac{H}{z_o}\right). \tag{A5}$$

Now, Eqs. (A2) and (A5) form a system of two equations with two unknowns, $N$ and $H$. However, there is no analytical

solution. Instead we must numerically search for a combination of $(N, H)$ that satisfies both conditions.

We do this by first using Eqs. (A2) and (A5) to directly calculate $r_e(H)$ and $\tau$ for narrowly spaced values of $N$, $H$, $c$, and $z_0$. The values used are given in Table A1. Let this table of values be known as lookup table 1 (LUT_1). Next, we create a second pre-calculated lookup table of $(N, H)$ given $r_e(H)$, $\tau$, $c$, and $z_0$. We shall call this LUT_2. For LUT_2 we use the same selection of values for $c$ and $z_0$, and evenly spaced values of $r_e$ and $\tau$ (see Table A1). For each $(r_e, \tau)$ combination at

given $c$ and $z_0$, we search LUT_1 for the combination of $(N, H)$ that minimizes the sum of the absolute percentage errors in $r_e$ and $\tau$. Finally, when performing inversions on MODIS observations of $r_e$ and $\tau$, we linearly interpolate LUT_2 to yield estimates of $N$ and $H$.

In Fig. (A1) we show the values of $N$ and $H$ retrieved by this method for a range of $r_e$ and $\tau$. Here we assume $z_0 = 500$ m with a temperature of 280 K and pressure of 900 hPA (corresponding to c = 0.002 g m$^{-4}$). We also show the difference in

retrieved $N$ and $H$ compared to assuming a fully adiabatic cloud. For some $(r_e, \tau)$ combinations, the differences are small. However, for clouds with larger $\tau$, the subadiabatic model yields deeper clouds with lower number concentrations. The differences in $H$ are greatest for clouds with large $r_e$, and the differences in $N$ are greatest for clouds with small $r_e$.

### Code availability

All code used to produce the results presented in this study is available from the Zenodo repository

(https://doi.org/10.5281/zenodo.7706791, Schulte, 2023).

### Data availability

All of the CloudSat and associated A-Train data used in this study are available from the CloudSat data processing center at cloudsat.cira.colostate.edu (last access: 7 March 2023). Other data necessary to reproduce the presented results are available on request.



## Author contributions

RS performed the data analysis and wrote most of the article. ML and JH helped conceptualize and focus the study, provided technical help and discussions, and helped edit the article.

## Competing interests

The authors declare that they have no conflict of interest.

## Acknowledgments

This work was funded by the National Aeronautics and Space Administration's *CloudSat* mission. The work of ML was performed at the Jet Propulsion Laboratory, California Institute of Technology, under a contract with NASA.

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

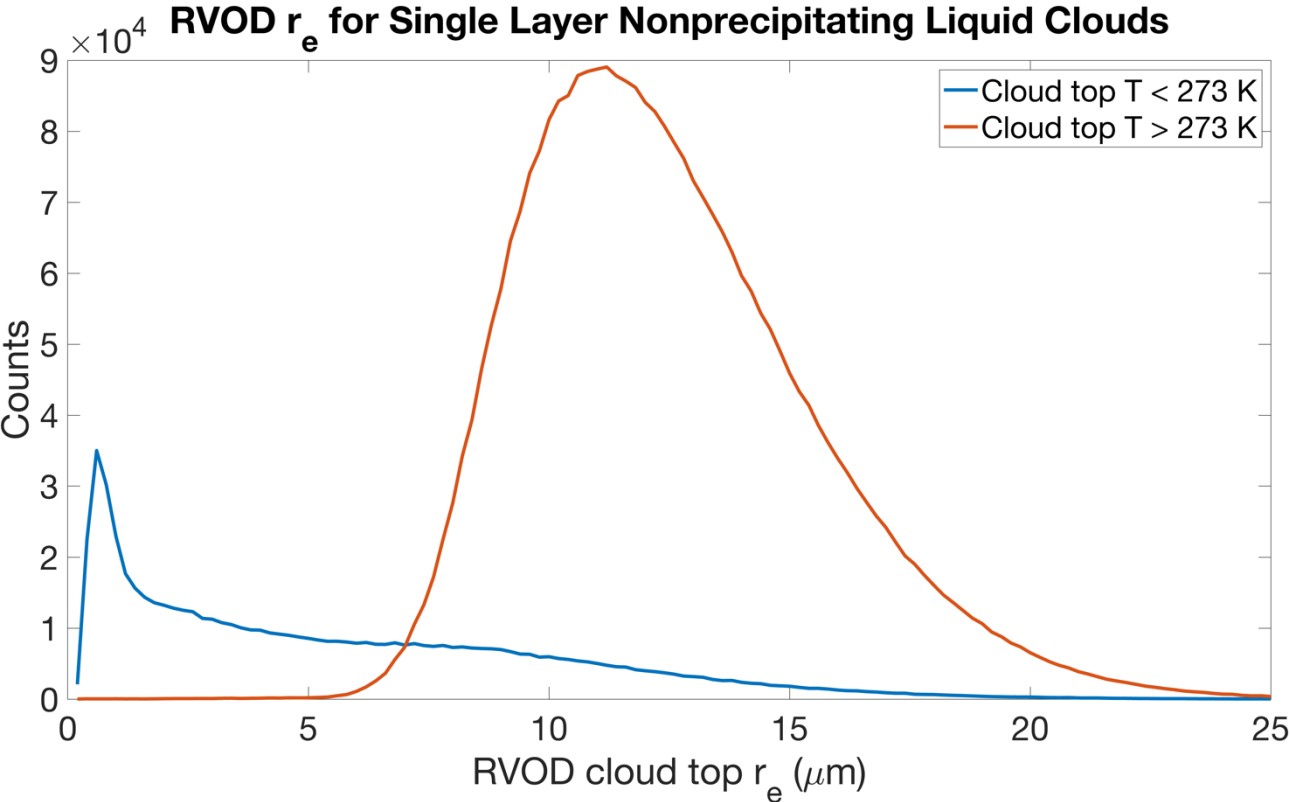

**Figure 1: Histograms of cloud top effective radius retrieved by RVOD, for cloud top temperatures less than or equal to 273 K (blue) or warmer than 273 K (red). CloudSat pixels from 2007-2016 that are classified as single layer liquid clouds from 2B-CLDCLASS-LIDAR and have no CPR reflectivities above -15 dBZ in the column are included. Cloud top height comes from 2B-CLDCLASS-LIDAR and the temperature at that height is taken from ECMWF-AUX.**



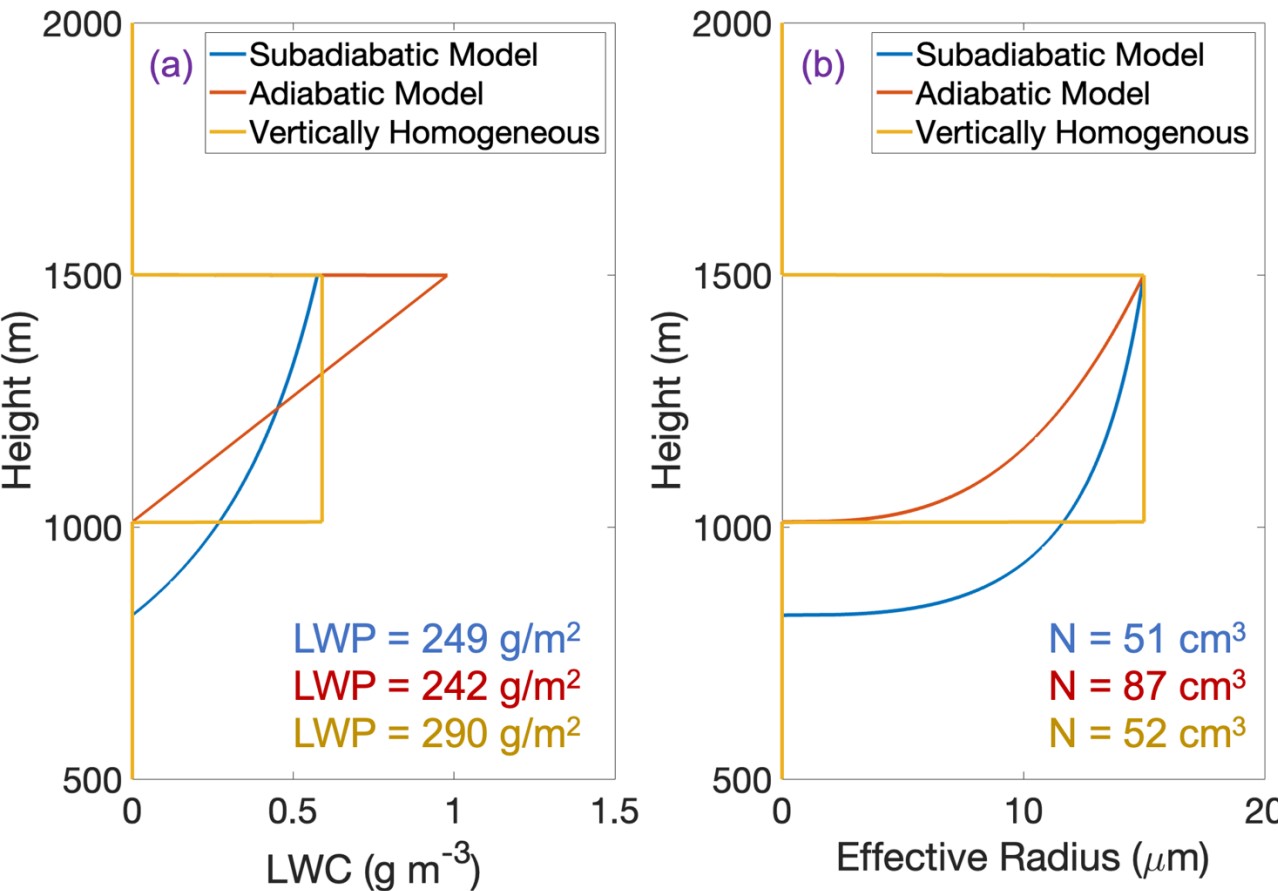

**Figure 2: Profiles of (a) liquid water content and (b) cloud droplet effective radius for a cloud with an optical depth of 29 and a cloud top effective radius of 15 $\mu$m. Each profile assumes a different vertical distribution of cloud water. The blue profiles use the subadiabatic model described in the text (with $z_0$ = 500 m), the red profiles assume adiabatic growth of cloud droplets from base to cloud top, and the gold profiles assume a vertically homogeneous cloud with the same cloud depth as the adiabatic cloud. LWP is the liquid water path of each cloud, and N is the cloud droplet number concentration (assumed constant throughout the cloud in each case).**



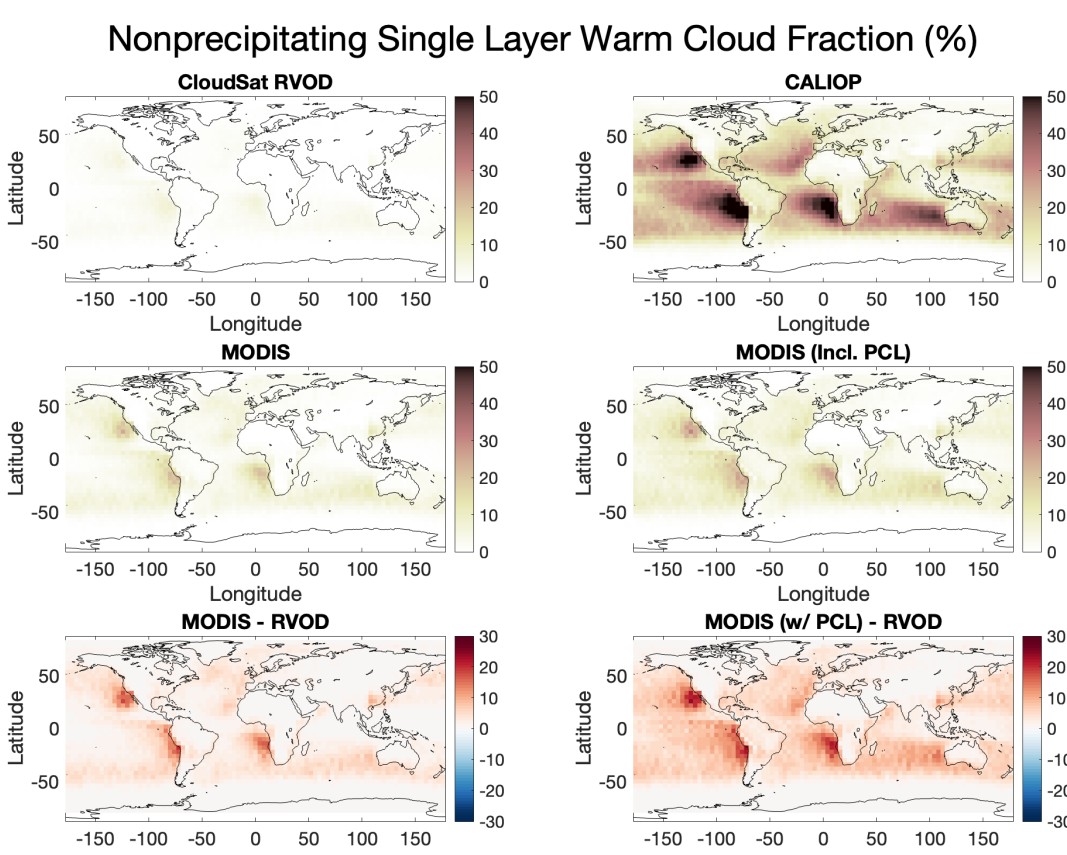

**Figure 3: Fraction of all CloudSat pixels that are identified as cloudy by RVOD, CALIOP, MODIS, or MODIS (including "partly cloudy" pixels), and that are further identified to be single layer nonprecipitating warm (SLNPW) clouds, according to the screening procedures laid out in the text. The bottom two panels show the difference in SLNPW cloud fraction between RVOD and MODIS (with and without partly cloudy pixels included).**



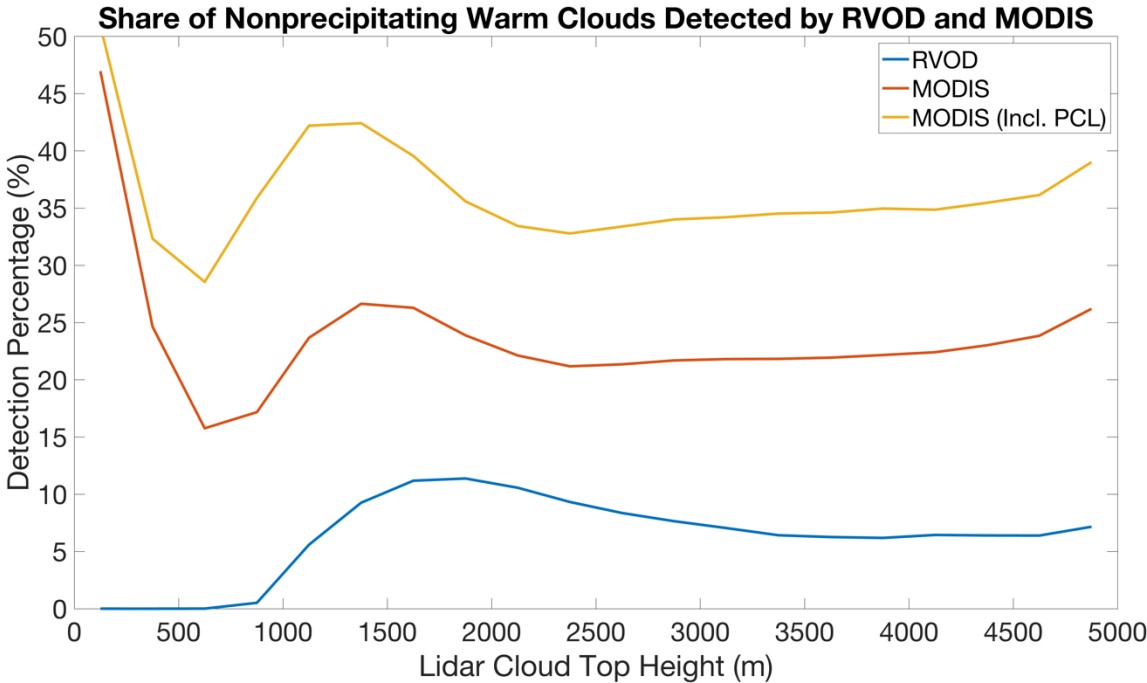

**Figure 4: Fraction of all CALIOP-detected single layer nonprecipitating warm clouds that are detected by RVOD, MODIS, or MODIS (including "partly cloudy" pixels), as a function of cloud top height.**



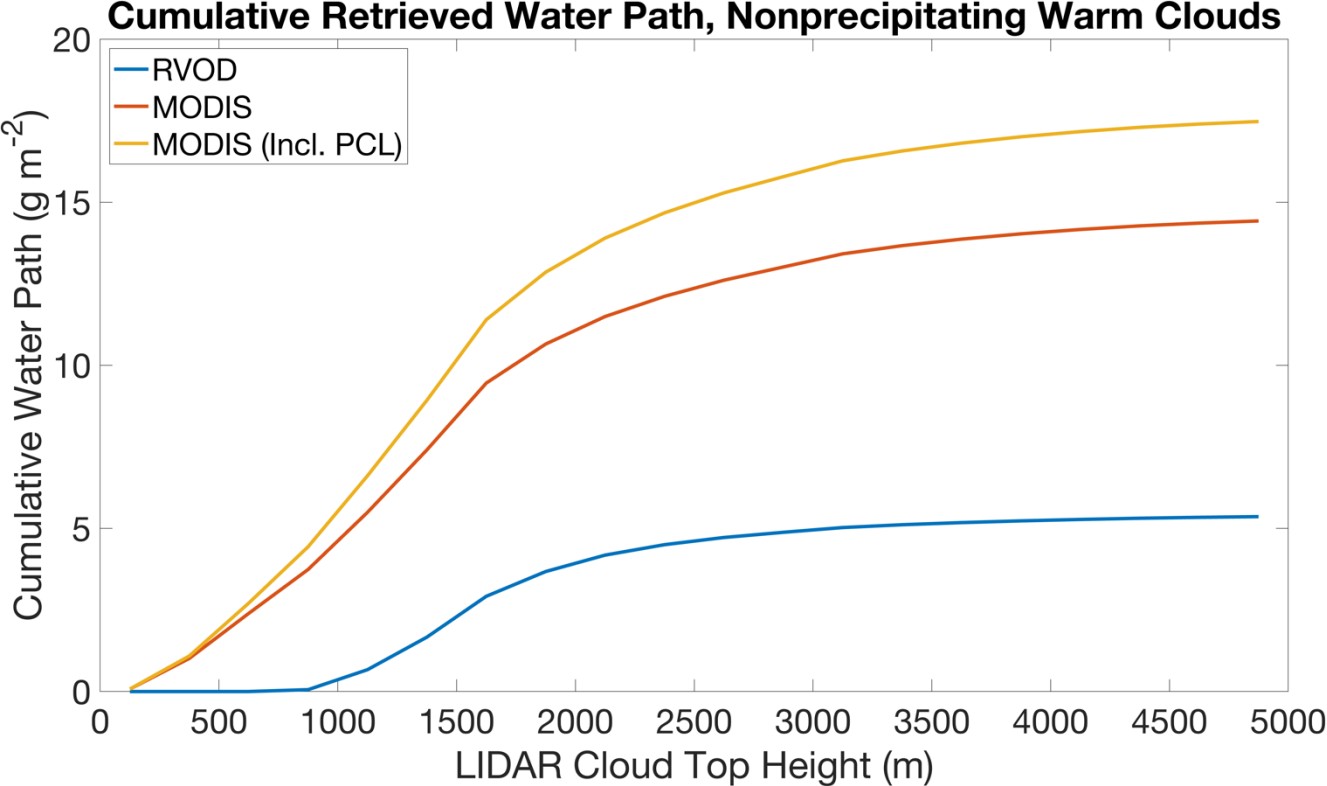

**Figure 5: Cumulative distributions of retrieved single layer nonprecipitating warm (SLNPW) cloud liquid water path, for RVOD,**
**MODIS, and MODIS (including "partly cloudy" pixels), as a function of cloud top height. The MODIS estimates come from our subadiabatic model. For each curve the numerator is the sum of the LWPs for all SLNPW pixels with cloud top heights up to the value given on the x-axis, while the denominator is always the total number of SLNPW pixels in the dataset.**



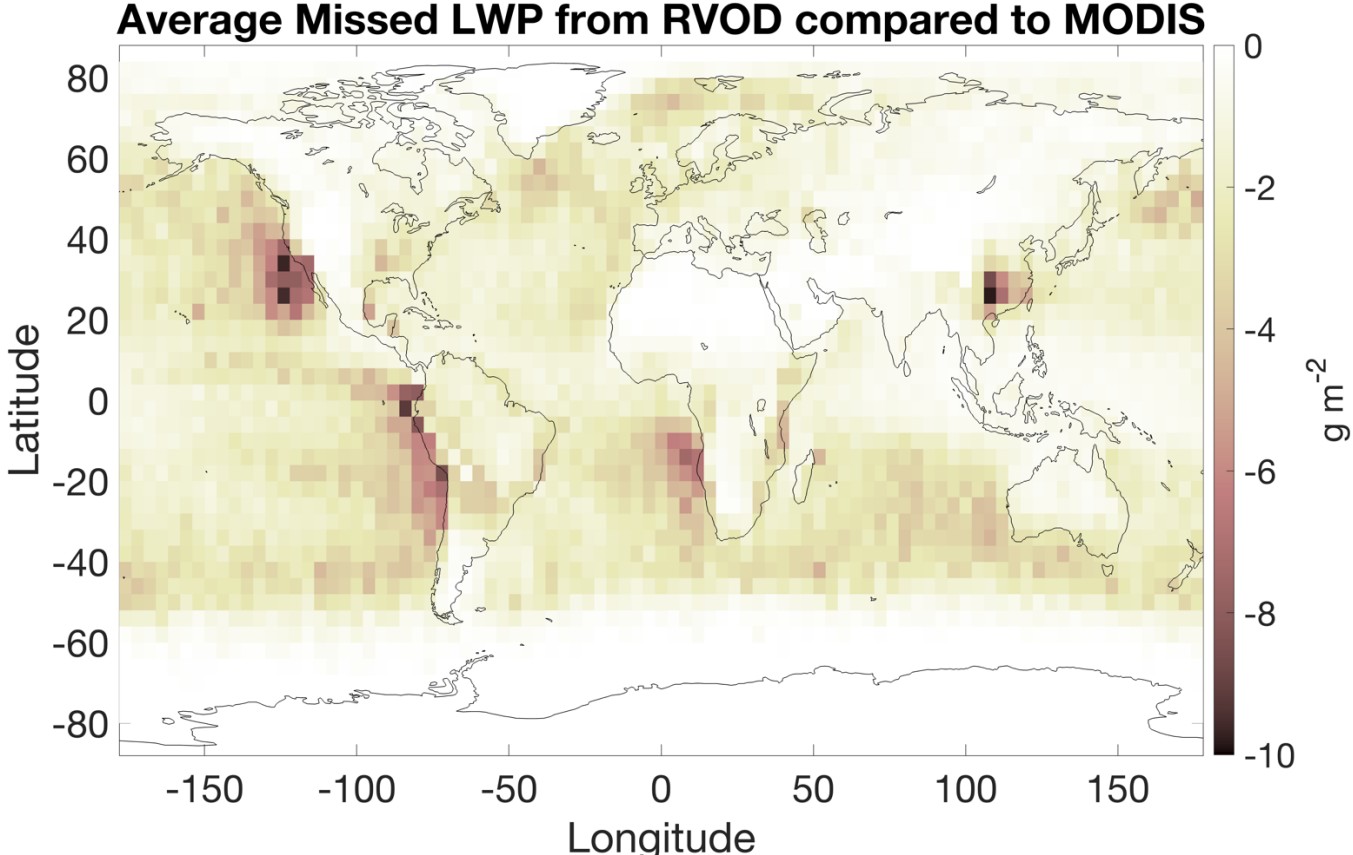

**Figure 6: Average single layer nonprecipitating warm cloud liquid water path (SLNPW CLWP) retrieved from the MODIS+CALIOP subadiabatic model, subtracted from the average SLNPW CLWP retrieved by RVOD. In both cases the denominator in the average is all CloudSat pixels (regardless of cloudiness), but only SLNPW clouds are considered in the numerator.**

**Figure 7:** (Left) Distributions of (a) cloud liquid water path, (c) column maximum cloud liquid water content, and (e) cloud top effective radius, as retrieved by either RVOD or our combined MODIS + CALIOP subadiabatic model, for all single layer nonprecipitating warm clouds with valid RVOD retrievals. (Right) Density plots comparing RVOD values of the three variables to the values retrieved from the MODIS + CALIOP model.



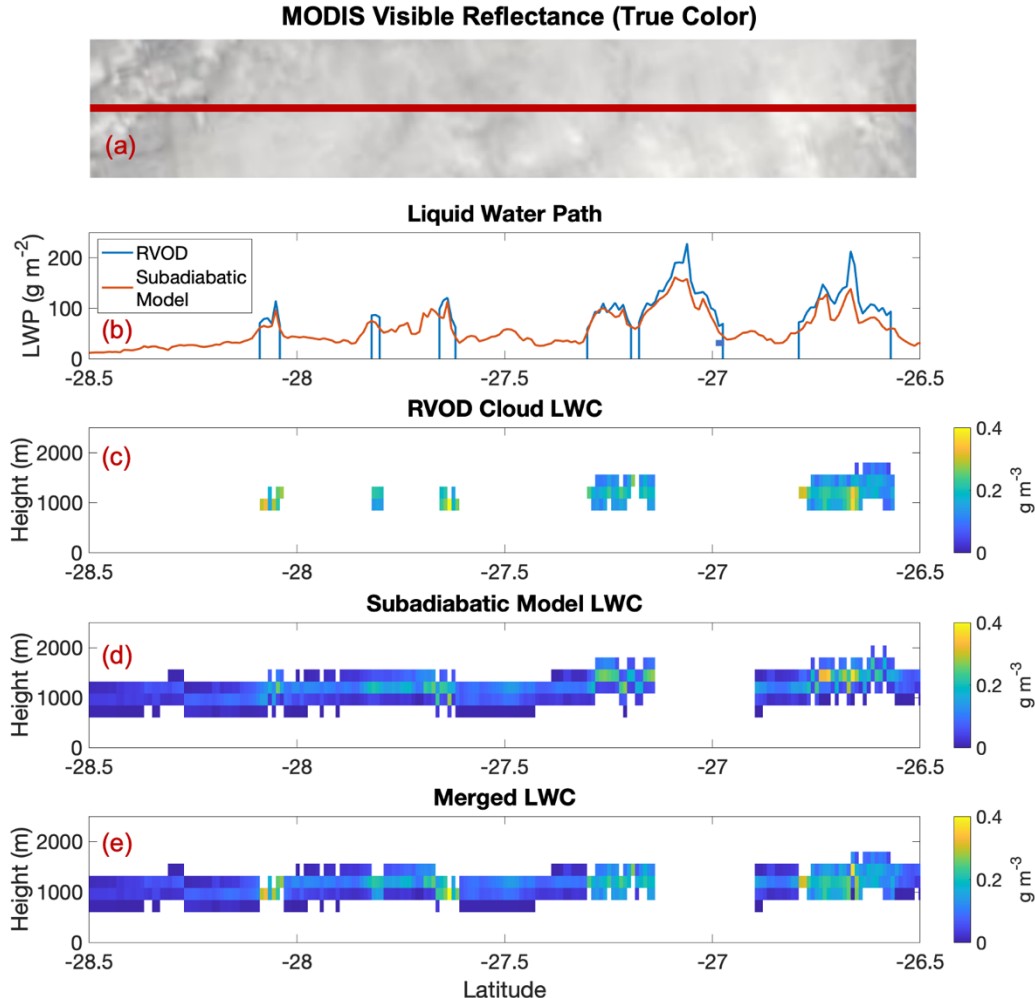

**Figure 8:** A nonprecipitating case study from 1 Feb 2007, CloudSat granule 4069. (a) MODIS visible imagery, with the CloudSat ground track overlain on top. (b) Retrieved liquid water path from RVOD (blue) and the MODIS subadiabatic model (red). (c) Retrieved liquid water content from RVOD. (d) Retrieved liquid water content from the MODIS subadiabatic model. (e) Merged
620 liquid water content, where the subadiabatic model is used for all pixels where RVOD has no retrieved liquid water.

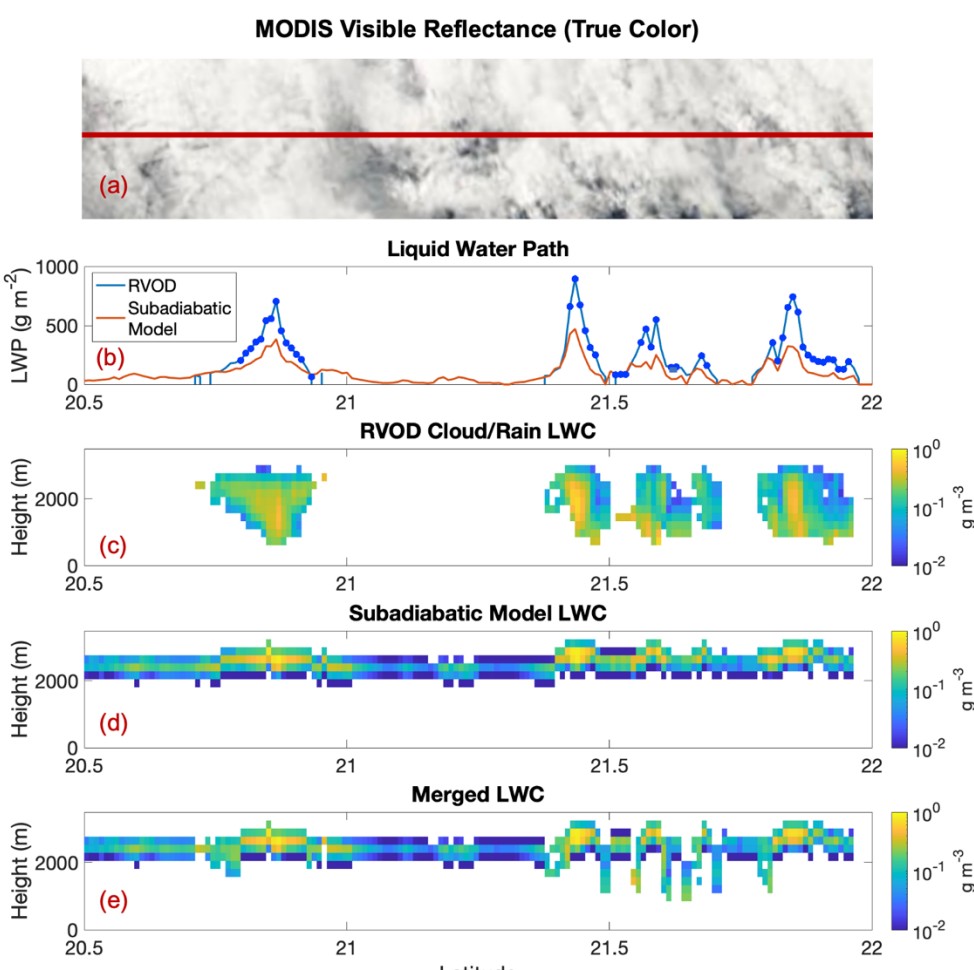

**Figure 9: A case study with precipitation, from 2 Jan 2007, CloudSat granule 3626. The panels follow the convention introduced in Fig. 8. In (b), the blue dots indicate CloudSat pixels for which the maximum reflectivity is greater than -15 dBZ (indicating the possibility of precipitation). Note that the RVOD liquid water content (Panel C) includes contributions from both the cloud and precipitation categories in the RVOD output.**

625



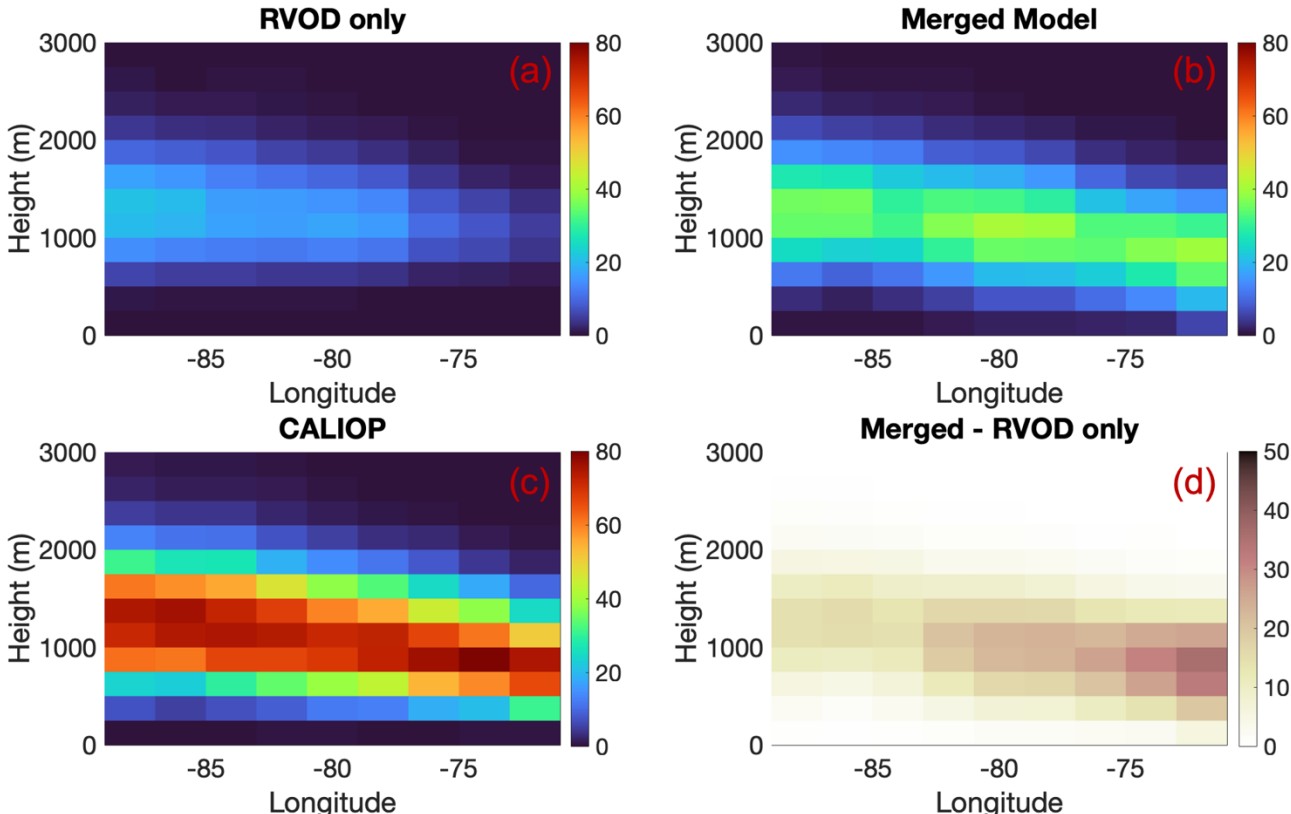

Figure 10: **Percentage of all pixels that are classified as nonprecipitating warm clouds by either RVOD (a), the RVOD/MODIS merged model (b), or CALIOP (c), for the VOCALS cross section defined in the text. The cloud fractions are stratified by longitude (in 2 degrees bins) and height (in 250 m bins). Panel (d) shows the difference between the merged model and RVOD.**



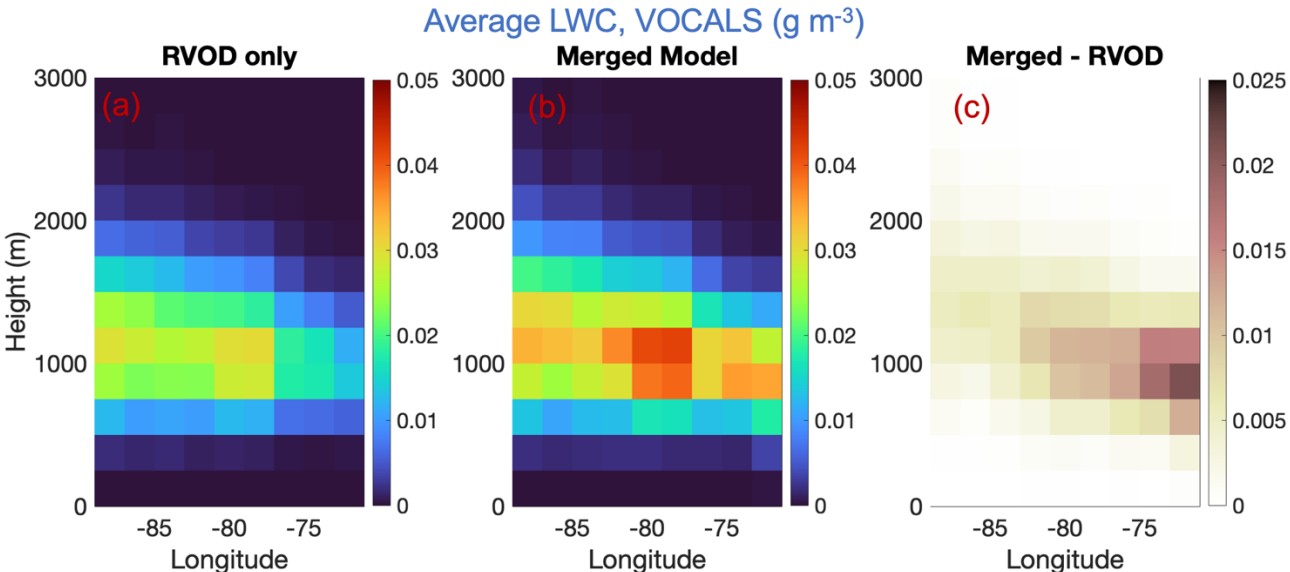

**Figure 11:** Average single layer nonprecipitating warm cloud liquid water content retrieved by (a) RVOD and (b) the RVOD/MODIS merged model, stratified by longitude and height for all observations of the VOCALS cross section. Panel (c) shows the difference between the merged model and RVOD.

| Experiment | SLNPW cloud detection % | Mean (SD) Effective Radius $r_e$ [$\mu$m] | Mean (SD) Cloud Thickness H [m] | Mean (SD) Number Concentration N [cm$^{-3}$] | Mean (SD) Liquid Water Path LWP [g m$^{-2}$] | Mean (SD) Column Max. Liquid Water Content LWC [g m$^{-3}$] |
|---|---|---|---|---|---|---|
| $\lambda = 3.7\mu$m, $z_0 = 500$m | 57.7 % | 12.1 (3.8) | 135 (84) | 167 (191) | 53 (62) | 0.16 (0.17) |
| $\lambda = 3.7\mu$m, $z_0 = 250$m | 57.7 % | 12.1 (3.8) | 151 (104) | 145 (164) | 54 (64) | 0.15 (0.16) |
| $\lambda = 3.7\mu$m, $z_0 = 100$m | 57.7 % | 12.1 (3.8) | 203 (175) | 104 (116) | 57 (67) | 0.14 (0.11) |
| $\lambda = 2.1\mu$m, $z_0 = 500$m | 53.6 % | 12.7 (4.2) | 139 (86) | 147 (154) | 56 (64) | 0.16 (0.17) |
| $\lambda = 2.1\mu$m, $z_0 = 250$m | 53.6 % | 12.7 (4.2) | 156 (108) | 127 (130) | 57 (66) | 0.16 (0.15) |
| $\lambda = 2.1\mu$m, $z_0 = 100$m | 53.6 % | 12.7 (4.2) | 211 (181) | 91.2 (92) | 60 (69) | 0.14 (0.11) |
| $\lambda = 1.6\mu$m, $z_0 = 500$m | 25.4 % | 13.2 (5.9) | 142 (92) | 184 (237) | 59 (68) | 0.18 (0.21) |
| $\lambda = 1.6\mu$m, $z_0 = 250$m | 25.4 % | 13.2 (5.9) | 160 (115) | 162 (208) | 60 (70) | 0.17 (0.18) |
| $\lambda = 1.6\mu$m, $z_0 = 100$m | 25.4 % | 13.2 (5.9) | 217 (192) | 121 (160) | 62 (73) | 0.15 (0.12) |

**Table 1:** Single layer nonprecipitating warm cloud detection percentage, plus the mean values of $r_e$, H, N, LWP, and column max LWC (with the standard deviations in parentheses), for each of the nine versions of the subadiabatic MODIS model described in the text. Each version uses a different combination of MODIS channel wavelength and scaling parameter $z_0$.





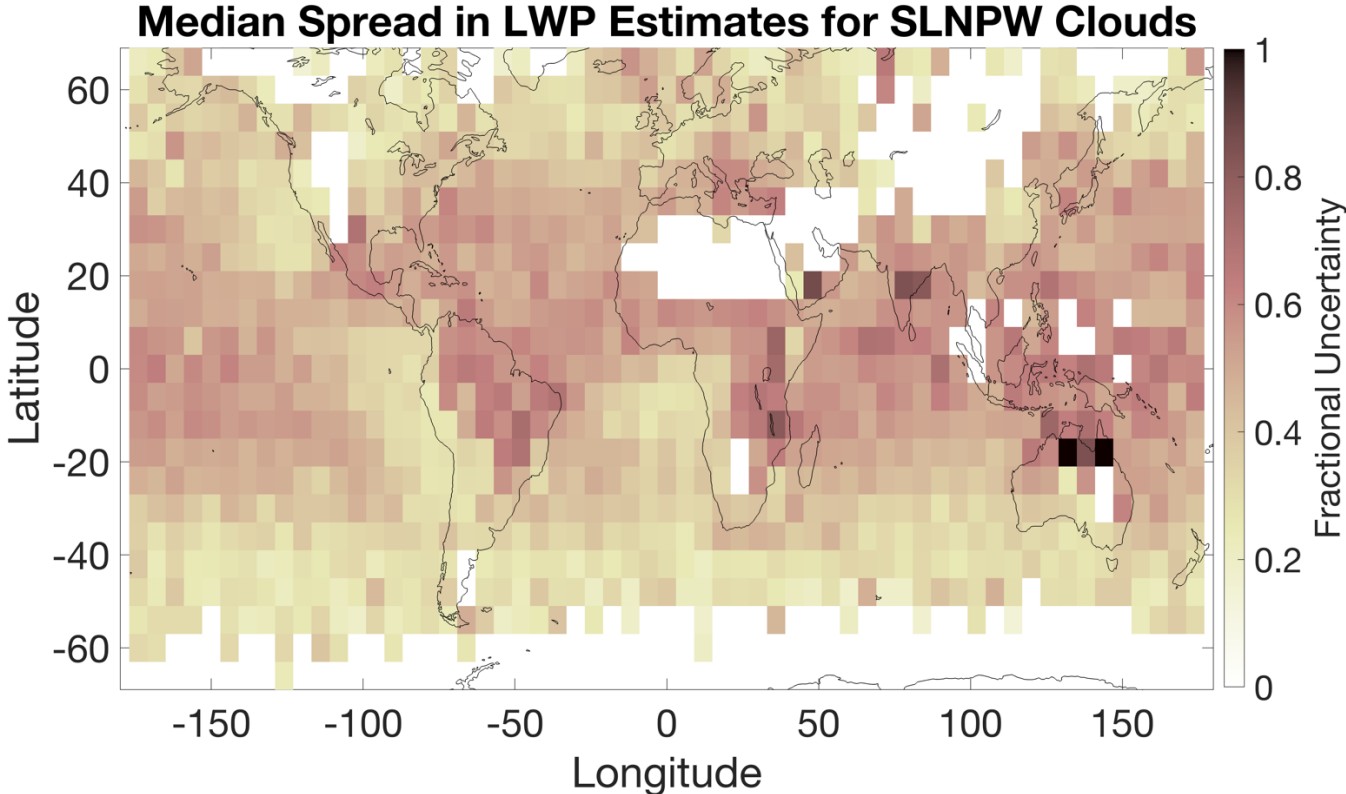

**Figure 12: Map of the median fractional uncertainty in the liquid water path estimate of all single layer nonprecipitating warm clouds contained within each latitude/longitude bin. The fractional uncertainty is based on 9 sensitivity tests using different MODIS channels and different values of $z_0$ and is described further in the text. Only bins with greater than 50 SLNPW pixels are included.**

| Parameter | Look-up Table(s) | Values Used |
|---|---|---|
| Droplet number concentration (N) | LUT_1 | 100 logarithmically spaced values from 1 to 10,000 cm$^{-3}$ |
| Cloud thickness (H) | LUT_1 | 10, 20, 30, …, 500; 550, 600, …, 5000 m |
| Cloud top effective radius ($r_e$) | LUT_2 | 2, 3, 4, …, 30 $\mu$m |
| Cloud optical depth ($\tau$) | LUT_2 | 60 logarithmically spaced values from 0.1 to 500 |
| Adiabatic condensation rate (c) | LUT_1, LUT_2 | 1.0, 1.2, 1.4, …, 4.0; 4.5, 5.0, …, 25 g m$^{-3}$ km$^{-1}$ |
| Scaling parameter ($z_0$) | LUT_1, LUT_2 | 50, 100, 150, …, 500; 750, 1000 m |

**Table A1: Values used to calculate the look-up tables LUT_1 and LUT_2. LUT_1 gives $r_e$ and $\tau$ as a function of N, H, c, and $z_0$, and LUT_2 gives N and H as a function of $r_e$, $\tau$, c, and $z_0$.**

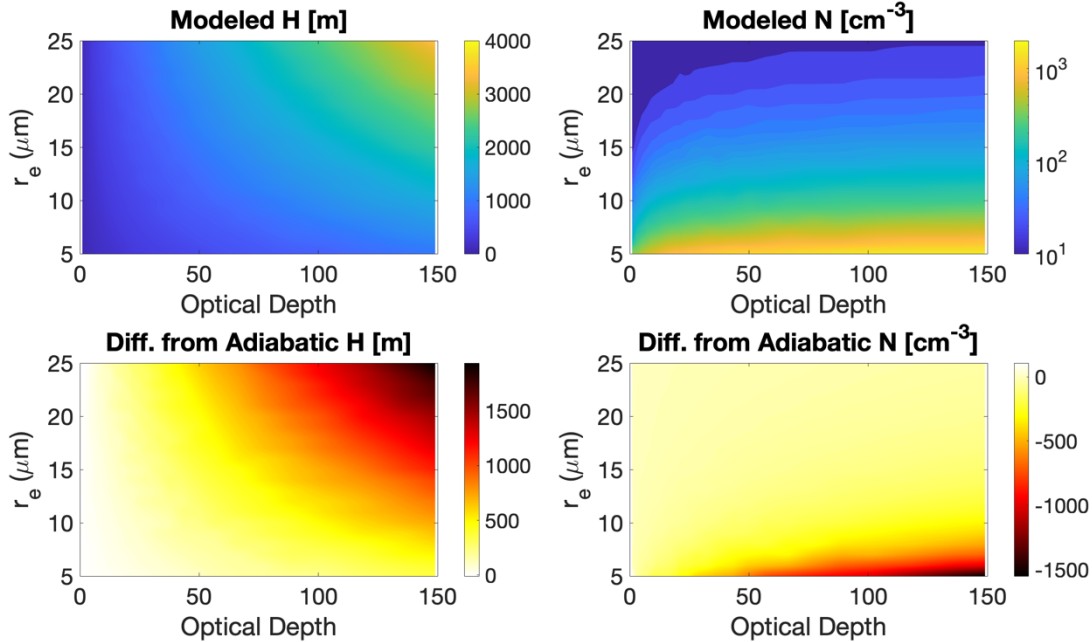

**Figure A1:** (Top) Cloud depth H (left) and droplet number concentration N (right) calculated by the subadiabatic cloud model for various combinations of optical depth and cloud top effective radius. (Bottom) Difference between the subadiabatic values of H and N and the values that would result from assuming a purely adiabatic cloud.
