# Peer review of "What CloudSat can't see: Liquid water content profiles inferred from MODIS and CALIOP observations"

_Atmospheric Measurement Techniques, 2023_

## Referee Comment (RC1)

Manuscript number: amt-2023-49

Full title: What CoudSat can't see: Liquid water content profiles inferred from MODIS and CALIOP observations

Author(s): Schulte et al.

CloudSat radar echoes from nonprecipitating liquid clouds can be either too weak or significantly contaminated by surface clutter effects. This paper found that the CloudSat 2B-CWC-RVOD products can miss these clouds by 73% and 84% compared to the overcast and partially cloudy pixels for liquid clouds detected by MODIS. In addition, this paper developed and validated a method to obtain the vertical profiles of cloud properties under such conditions using MODIS and CALIOP products collocated with CloudSat. The method aims at filling the data gap in nonprecipitating liquid clouds that the CloudSat 2B-CWC-RVOD product misses. The method uses the sub-adiabatic assumption to interpret the MODIS-derived cloud properties into the vertical profiles of these properties. The developed method can successfully reproduce the probability density distribution of the cloud water path for clouds detected by radar, confirming the consistency with that from the CloudSat 2B-CWC-RVOD product.

Furthermore, case studies demonstrated that the cloud profiles based on the new method could supplement the CloudSat profiles. Thus, this method is considered a plausible approach to obtaining the vertical profile of nonprecipitating liquid clouds when the CloudSat radar profile is unavailable. This paper is well-written and well-organized. The topic is suitable for a formal publication in Atmospheric Measurement Techniques. This manuscript can be accepted after minor revisions. I have only several comments that the author can consider for improving the manuscript.

Minor comments

1. Eq. (11) on page 7: The cloud depth $H$ (i.e., geometric thickness) is derived from cloud optical thickness and effective radius from the MODIS cloud product. I wonder if this formulation guarantees that the cloud bottom does not reach the

ground level or does not provide such situations in practice.

2. Lines 191-192 on page 8: As the MODIS product assumes the single-layer homogeneous cloud in the retrieval process, applying this assumption to vertically inhomogeneous clouds leads to a systematic bias in the retrieval products (i.e., $\tau$ and $r_e$) due to vertically inhomogeneous microphysical properties (Platnick, 2000). Although these biases are small for adiabatic clouds (Saito et al., 2019), it would be good to mention this here.

3. Line 239 on page 8: "estimates of estimates of" should be "estimates of."

4. Line 281 on page 10: Figure 9a shows SW reflectance (true color), but the corresponding description indicates IR brightness temperature. Please clarify this.

5. Line 290-291 on page 10: "are some are some" should be "are some."

Platnick, S. (2000). Vertical photon transport in cloud remote sensing problems. Journal of Geophysical Research, 105(D18), 22,919–22,935. https://doi.org/10.1029/2000JD900333

Saito, M., Yang, P., Hu, Y., Liu, X., Loeb, N., Smith Jr, W. L., & Minnis, P. (2019). An efficient method for microphysical property retrievals in vertically inhomogeneous marine water clouds using MODIS-CloudSat measurements. Journal of Geophysical Research: Atmospheres, 124. https://doi. org/10.1029/2018JD029659

---

## Author Response (AR1)

**Reviewer #1 Comments**

1. Eq. (11) on page 7: The cloud depth H (i.e., geometric thickness) is derived from cloud optical thickness and effective radius from the MODIS cloud product. I wonder if this formulation guarantees that the cloud bottom does not reach the ground level or does not provide such situations in practice.

In rare cases, the method for retrieving H described in the appendix results in a cloud depth that is greater than the cloud top height – that is, physically impossible. In these cases, we iteratively increase the value of the condensation rate (c) by 1% and re-compute H, repeating this process until we arrive at a value of H that is less than the cloud top height. We have added this explanation to the revised manuscript (Lines 416-418).

2. Lines 191-192 on page 8: As the MODIS product assumes the single-layer homogeneous cloud in the retrieval process, applying this assumption to vertically inhomogeneous clouds leads to a systematic bias in the retrieval products (i.e., t and re) due to vertically inhomogeneous microphysical properties (Platnick, 2000). Although these biases are small for adiabatic clouds (Saito et al., 2019), it would be good to mention this here.

This is a good point. We have mentioned this in Lines 193-196 and now cite both Platnick 2000 and Saito et al. 2019.

3. Line 239 on page 8: "estimates of estimates of" should be "estimates of."

Fixed as suggested.

4. Line 281 on page 10: Figure 9a shows SW reflectance (true color), but the corresponding description indicates IR brightness temperature. Please clarify this.

The figure description should say true color reflectance. This has been corrected in the revised manuscript (Line 271).

5. Line 290-291 on page 10: "are some are some" should be "are some."

Fixed as suggested.

**Reviewer #2 Comments**

Line 210: The SLNPW cloud fraction shown in Figure 3 represents the SLNPW cloud fraction of CloudSat pixels, not CALIOP or MODIS cloud fraction. The title of Figure 3 and the text in the manuscript could be modified to explicitly state that the cloud fractions of MODIS or CALIOP are CloudSat pixels-based cloud fractions. Additionally, including the actual MODIS and CALIOP SLNPW cloud fractions (or the total cloud fraction), would provide the reader with a comparison of MODIS and CALIOP cloud fractions to the CloudSat pixels-based cloud fraction.

This is a good clarification, and we have adjusted the text (Lines 210-211; 214-215) and the title of Figure 3 to clearly state that what is shown is the fraction of CloudSat pixels with clouds, not cloud fraction at CALIOP or MODIS resolution. Because the SLNPW classification as we have

defined it is only possible at the CloudSat pixel level, we have refrained from adding any additional plots of MODIS or CALIOP cloud fraction.

Line 258: The MODIS-based sub-adiabatic LWP estimations show a positive bias in Fig. 7 panel (d), which could be attributed to the COT biases in the MODIS observations. Perhaps the authors could elaborate on the criteria used to choose the best COT observations from MODIS (e.g., flags) and consider the solar zenith angle when choosing MODIS COT observations.

The LWP estimates (in panel b) are actually almost total unbiased with respect to the RVOD estimates. What panel (d) shows is that the MODIS-based maximum LWC (i.e., the LWC in the radar bin at or near the top of the cloud) is positively biased. This has to do with the tendency of the subadiabatic model to make clouds that are thinner (with a steeper growth rate in LWC) than those retrieved by RVOD, as discussed in Lines 259-264. Nevertheless, it is true that MODIS COT observations have been shown to be biased at high solar zenith angles. We plan to implement a correction for this (as suggested in Lebsock and Su 2014, doi: 10.1002/2014JD021568) in the next reprocessing of the algorithm. However, as the focus in this paper is on warm clouds, most of the pixels considered are at low latitudes and so any solar zenith angle effects are small.

Line 267: The title of Figure 8 panel (a) says it is MODIS True color, but the text states it is the 11um brightness temperature.
MODIS true color is correct. The figure description in the text (Line 271) has been updated.

Line 65: Is -> It

Fixed as suggested.